# Multi-Classifier of DDoS Attacks in Computer Networks Built on Neural Networks

**Andrés Chartuni and José Márquez ***

Computer Science and Engineering Department, Universidad del Norte, Barranquilla 080020, Colombia; achartuni@uninorte.edu.co

*** Correspondence: jmarquez@uninorte.edu.co

**Featured Application: Authors are encouraged to provide a concise description of the specific application or a potential application of the work. This section is not mandatory.**

**Abstract:** The great commitment in different areas of computer science for the study of computer networks used to fulfill specific and major business tasks has generated a need for their maintenance and optimal operability. Distributed denial of service (DDoS) is a frequent threat to computer networks because of its disruption to the services they cause. This disruption results in the instability and/or inoperability of the network. There are different classes of DDoS attacks, each with a different mode of operation, so detecting them has become a difficult task for network monitoring and control systems. The objective of this work is based on the exploration and choice of a set of data that represents DDoS attack events, on their treatment in a preprocessing phase, and later, the generation of a model of sequential neural networks of multi-class classification. This is done to identify and classify the various types of DDoS attacks. The result was compared with previous works treating the same dataset used herein. We compared their classification method, against ours. During this research, the CIC DDoS2019 dataset was used. Previous works carried out with this dataset proposed a binary classification approach, our approach is based on multi-classification. Our proposed model was capable of achieving around 94% in metrics such as precision, accuracy, recall and F1 score. The added value of multiclass classification during this work is identified and compared with binary classifications using the models presented in the previous.

**Keywords:** computer networks; data preprocessing; DDoS attack; machine learning; neural networks

## 1. Introduction

Computer network systems have been deployed to achieve interconnectivity between devices and carry out essential business tasks. However, this has created great dependence on the main functions of an entity regarding its connection systems [1]. Main areas such as banks, health entities and service providers are exposed to risks of instability because of their strong and necessary dependence on computer networks. Because of this dependency, it is essential to keep networks in an optimal state, specifically to maintain connectivity, performance, and security. Network performance can be strongly affected by a security failure, causing instability to the point of network inoperability [2].

To generate an anomaly in the network, different types of attacks have been used. Among these, one of the main ones is the denial of service (DoS) attack. There has been an increase in the number of such attacks in the last ten years, establishing them as a significant threat to the stability of networks resulting from the alteration of various services [3].

A very common deployment mode of this attack is better known as distributed denial of service (DDoS). This deploys a DoS attack simultaneously across a computer network. A successful DDoS attack results in the exhaustion of bandwidth, routing device processing, network or processing resources, memory, database, and bandwidth of server input and output operations [4,5].

There are measures to prevent these attacks. However, it is important to identify the attack characteristics to take optimal actions to avoid its recurrence [6]. Unfortunately, the characteristics of these attacks can vary considerably, so it is important to be able to identify the type of alteration.

Network systems are in constant transformation, making dataset age an important attribute at the time of their selection. Different datasets have been proposed. First are DARPA'98 and '99 [7], which have categorized information for intrusion detection. However, their implementation dates are very old and there is a risk of not representing current information correctly. On the other hand, there are the datasets provided by CAIDA, which were captured with the most recent date in the year 2019. However, the information is not categorized and cannot be used for supervised learning in ML [8].

The Canadian Institute for Cybersecurity (CIC) has generated a dataset called CIC DDoS2019 [9]. This dataset was produced in 2019 and has around 25 GB of labeled information, making it an ideal source of information for DDoS attack detection work. Furthermore, the reason why we opted for CIC DDoS2019 dataset is due to its high reliability and its similarity to real world attacks [9].

Different ML detection proposals using the CIC dataset have been used to detect DDoS attacks, obtaining excellent results [10,11]. However, their detection models operate with the training of one attack per model, resulting in a binary classification of flow detection, i.e., benign and malignant.

This work proposes the implementation of a sequential ML model, a data processing framework and its subsequent processing, which allow detection at a multiclass level of attacks instantiated in the CIC dataset [9].

Previous works using the CIC DDoS2019 dataset are based on binary classification. Our approach's novelty is the implementation of a multi-classification model. This will allow for a better solution in the scenario when a malign flow is detected due to the model's capability of outputting the type of attack present in the flow. Furthermore, we propose to combine labels resulting into one label that succeeds at classifying two types of attack given their similarities.

The paper presents a section of previous work that describes and analyzes advances related to the detection of DDoS attacks using ML techniques. This is followed by a methodology section, which explains the various procedures performed during the research. These steps include dataset selection, information preprocessing, and definition of the model used. The third section includes an evaluation of the defined model and the results. Finally, future work and conclusions of the research are addressed.

## 2. Previous Works

This section describes the latest work and progress in detecting DDoS attacks, as well as a description of the deployments and data used to obtain the published results. He et al. [12] obtained data from a simulation of four types of attacks: SSH brute force, ICMP flooding, DNS reflection, and TCP SYN. Network traffic is generated by virtual machines and in turn a monitoring system captures this traffic, which later feeds a dual ML system. That system is made up of two layers. The first consists of an already trained model and the second of a process that feeds that model at runtime to adjust it to values evaluated during execution.

The ML system proposed by He et al. [12] is composed of a set of different methods applied to the information collected. Each method is tested against a set of four attacks generated and scores are compared between those methods. The methods applied by He et al. were linear regression (LR); support vector machines (SVM; linear kernel, radial basis function and poly); decision tree, naive Bayes (NB); random forest (RF), unsupervised K-means and Gaussian expectation-maximization.

The results show that the best method of the compendium used by He et al. [12] for the dataset generated is random forest because of its high precision during the evaluation stage.

Aamir et al. [13] proposed a framework based on four main phases: obtaining the dataset, feature engineering, evaluating the ML model, and results. The dataset is obtained by systematically exploring published and validated datasets that contain evidence of DDoS attacks. Once the dataset has been selected, feature engineering is carried out. In the feature engineering stage, the dataset is analyzed in order to recognize its environment and perform tasks of duplication and collinearity identification between different attributes. Similarly, an adjustment is made to make it suitable for training the selected ML model.

Model evaluation includes a first stage of training, adjustment of hyperparameters based on the results, and evaluation of the resulting model following its modifications.

Aamir et al. [13] evaluated five machine learning models: SVM, RF, artificial neural network (ANN), NB, and knearest neighbors (KNN). The author highlighted the importance of the processes before deployment of the ML models and attributed the results of precision, false positives and recall of the different methods to the established work framework. Adjusting the hyper-parameters allows the model designer to fine tune certain settings which, depending on the qualities of the data, can significantly impact the model's result.

Elyased et al. [10] used the same dataset as the present investigation, i.e., that proposed by the authors of the CIC [9]. This dataset is the same one used in our research. That work implemented a method based on deep learning structured with a recurrent neural network (RNN) and an autoencoder.

The method of Elyased et al. contains a data preprocessing stage in which certain network attributes are not considered for model training. The training phase is carried out to obtain a binary classification model. The model is trained in order to classify whether a specific attack flow is malignant or benign by dividing the dataset by attack type and feeding the model in this way.

The model is composed of an input layer and then, in the hidden layer, a RNN is implemented with an autoencoder, ending with an output layer in which the binary classification is performed. In the autoencoding phase, there is a reduction in the dimension of the input information to then perform decoding, in which the coded attributes are reconstructed to generate an output in the final layer that assimilates information belonging to the label of the registry [14].

The result of the above research is a model called DDosNet that has high accuracy in classifying benign and malignant flows. In their section on future works, Wang et al. stressed that although it is important to identify whether the flow is malignant or benign, it is also desirable to be able to detect the specific type of attack evidenced by the flows. This implementation is proposed as work to be done in new DDoSNet implementations.

Li [11] used the CIC dataset, the same dataset used in our research. Li proposed the evaluation of three models, which use concepts such as dense neural networks, self-encoders, and Pearson's correlation coefficient (PCC). Dense neural networks are made up by layers whose neurons are fully connected be- tween each layer, i.e., each neuron in each layer receives information from all neurons in the previous layer and sends information to all neurons in the next layer [15]. In this case, the PCC attempts to measure linear correlation between two attributes of the dataset. Once PCC is applied to the attributes, a value is obtained between −1 and 1. There is no linear correlation if the value is 0, 1 if the linear correlation is totally positive, and −1 if there is totally negative linear correlation [16].

In Li et al.'s work, five major processes were executed: analysis and engineering of dataset attributes, dataset training, adjustments based on results of the training, testing with information not previously seen, and comparison with traditional ML methods. The three resulting models perform a binary classification of the flow, i.e., like Elyased et al. [10], the model is limited to binary classification as benign or malignant. The composite models can be understood as a disaggregation of the proposed steps. That is, the initial model is a dense neural network, the second a dense neural network with an implemented autoencoder, and the third a dense neural network with an autoencoder that receives information subsets as

input data which through the PCC analysis does not include attributes that have strong linear correlation.

In the third model from Li et al., the training is executed with a subset of the dataset that only contains one type of attack and based on the PCC analyses, the attributes of the subset that contain a PCC value greater than 0.9 and less than −0.9 are eliminated.

The three proposed models give different results depending on the attack classification as malignant or benign. In general, the model constructed only with dense neural networks generally attains the best score among the three proposed models, followed by the dense neural network with autoencoding and, lastly, the dense neural network with autoencoder and PCC analysis.

The limitations of the existing works relay on the classification score and its behavior on each label. The models' score tends to variate considerably between the labels with a model's scores varying between the 0.5 and 0.9 on average.

## 3. Methodology

### 3.1. Dataset Selection

Datasets constitute a main element in the ML field. When initiating any type of conceptualization of a ML model, it is essential to obtain or develop a dataset that assimilates realworld situations or events to be classified in order to conduct successful training and a validation phase [17].

For the choice of dataset in our research, three sources of information were considered: "DDoS Attack 2007" [18]; "Anonymized Internet Traces" [19] provided by CAIDA; "CICDDoS2019" [9] provided by CIC.

The two remaining datasets have very complete information on network traffic captures and both provide a compilation of information exceeding 20 GB. Therefore, the amount of information is more than enough for ML model training. However, the data from the Anonymized Internet Traces dataset lacks preprocessing and therefore labels in its records. Consequently, training a model like the one used in the present work with this information would involve an arduous phase of data labeling.

Based on the assessment and analysis of the datasets explored, we decided to use the CIC dataset for training and validation of the model used.

The dataset provided by CIC is designed to provide reliable information for training DDoS attack detection models. Unlike previously proposed datasets, CIC seeks to remedy drawbacks of these, such as the lack of categorized information, attackers with implementation of simple-level intrusions, and a lack of generation of modern attacks such as NTP, NetBIOS, and TFTP [9].

The attacks implemented in the CIC dataset were classified into two main branches, Reflection-based DDoS and Exploitation-based DDoS.

- Reflection-based DDoS is a type of attack in which the attacker remains hidden during its execution through the legitimate use of third-party components. The attacker sends packets to reflecting servers with the source IP configured as the victim's IP, thereby overloading the victim with responses from these servers. The attacks evidenced in this dataset can be subcategorized as TCP (MSSQL, SSDP), UDP (CharGen NTP, TFTP), or TCP/UDP (DNS, LDAP, NetBIOS, SNMP, PortMap). That is, they can be performed using both protocols [9].
- Exploitation-based DDoS, in concept, are similar to Reflection-based DDoS with the difference that these attacks can be conducted through application-layer protocols using transport-layer protocols. The attacks evidenced in this dataset can be subcategorized as TCP (Syn-Flood) or UDP (UDP Flood, UDP-Lag) [9].

The taxonomy of attacks proposed by Sharafaldin et al. [9] is illustrated in Figure 1.

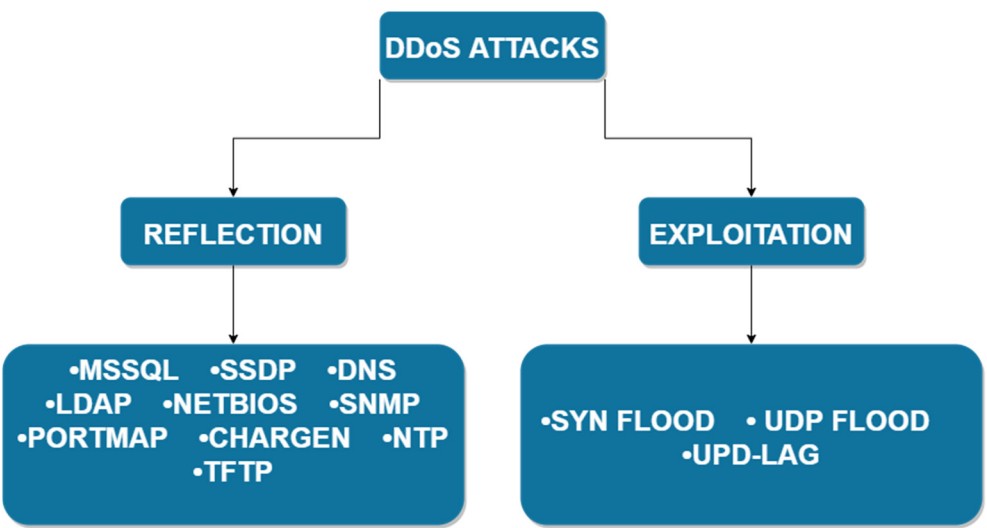

**Figure 1.** DDoS attack taxonomy [8].

The dataset is available in two formats, packet capture (PCAP) and comma-separated values (CSV).

The PCAP files contain raw information captured by the network monitoring tool and are not categorized. However, a table is provided that specifies the period of the attacks. On the other hand, CSV files are categorized and processed to record in rows the flow of information over the network. These files were generated from the PCAP files using the CICFlowMeter [20] tool, which consists of the generation of traffic flow across the network.

Model training and evaluation was mainly done using the files already preprocessed and offered by CIC. Based on information about the quality of these files, as set out by Li [11] in reference to obtaining fewer invalid values (NaN, and infinite) in the new files generated by CICFlowMeter, these new files generated by CICFlowMeter are in the CSV format. However, this did not improve the accuracy of our model.

The total amount of information in the dataset is 28.1 GB in CSV extension files and 19.1 GB in PCAP extension, considered sufficient for model training and validation [21].

The CIC DDoS dataset contains 87 attributes, of which 83 were generated by the CICFlowMeter application and four instantiated by Sharafalding et al. [9]. However, network attributes were eliminated so the model can be deployed in future work in another network structured differently than that in which the CIC data were captured. In total, 78 attributes were used. For an in-depth look at these attributes, refer to Sharafaldin et al. table of attributes [9].

### 3.2. Data Preprocessing

It is possible to train the model with the data after removing certain unwanted attributes, as mentioned above. However, deployment of a data preprocessing phase results in more robust training and thereby a more accurate model [22].

First, each file belonging to the dataset is unified into an information instance. Each file contains mostly one type of attack. Because the main objective is the implementation of a model capable of classifying whether there was a benign or malignant flow and if the flow is malignant to output the name of the attack, it is necessary to feed the model with information that considers different labels.

The unified dataset contains values invalid for training. There are inputs of NaN and infinite values, which are discarded to execute model training. About 3% of the data were removed in this procedure.

The objective with the model is multiclass classification in the context of DDoS attacks. To do this, an encoding procedure is necessary. For this procedure, it is necessary to transform the dataset labels so they can be used by the model. In our research, we used

One Hot Encoder (OHE). OHE processing assigns a new column for each label represented in the dataset and a value of 1 if the record belongs to this category or 0 otherwise [23]. The need for this encoding is based on the implementation of an output layer with a Softmax activation function. The Softmax function transforms a vector of k real values to one of k real values that add to unity [24].

In the next step of the data preprocessing, the data are normalized based on the L2 normalization. The use of L1 normalization was contemplated but results in the training phase showed small values of the model's accuracy metric. The L2 standard was applied to each column. That is, the attributes of the dataset, with $x$ being each occurrence of a record, define the normalization Formula (1) as:

$$||x||_2 = \sqrt{\sum_{i=1}^{n} |x_i|^2},$$

(1)

Normalizing the dataset records generally results in much faster training [25]. This behavior is evidenced in our model. In addition to improvement in convergence during the training phase, this normalization yielded better model accuracy owing to the handling of a uniform range of dataset attributes.

We conducted transformation by quartiles after the data normalization. This method transforms the attribute values of each dataset input in order to follow a uniform or normal distribution. For the model used, we decided to follow a normal distribution because of improvement in the accuracy metric compared to the uniform distribution.

In analyzing the information of the unified dataset, an imbalance of classes was detected, which resulted in a classification biased to a certain class because of its high frequency in the dataset. Table 1 presents the occurrence count according to the classes in the dataset.

**Table 1.** Distribution of labels in the loaded dataset.

| Tag | Occurrences | Percentage |
|---|---|---|
| TFTP | 975,826 | 37.605% |
| DrDoS_SNMP | 257,240 | 9.913% |
| DrDoS_DNS | 245,654 | 9.467% |
| Syn | 228,521 | 8.806% |
| DrDoS_MSSQL | 220,052 | 8.480% |
| DrDoS_NetBIOS | 198,815 | 7.662% |
| UDP | 159,751 | 6.156% |
| DrDoS_SSDP | 128,614 | 4.956% |
| DrDoS_LDAP | 107,084 | 4.127% |
| DrDoS_NTP | 59,736 | 2.302% |
| Portmap | 8831 | 0.340% |
| BENIGN | 4754 | 0.183% |
| UDPLag | 91 | 0.004% |

As seen in Table 1, the TFTP class has a strong presence in the dataset, whereas labels of the BENIGN or UDPLag type had small percentages of occurrence. Therefore, balancing the dataset is necessary for better training of the model.

In this research we propose the use of a fully balanced dataset, i.e., an equivalent percentage between the classes. There are three ways of balancing: duplicity of the labels to be balanced; elimination of label occurrences with the highest percentage in the dataset; a synthetic minority oversampling technique (SMOTE) [26].

The first balancing method resulted in a successful training phase. However, in the test phase, we validated a case of overfitting in the dataset used for training, because the accuracy values of the predictions were small compared to those obtained in the training phase.

The second method, eliminating records belonging to strongly represented labels in the dataset, was applied and tested. In this case, the training phase produced low model precision. We conclude that the results of this technique are attributable to a strong reduction of the dataset in order to find class balance. SMOTE was applied in our model as a class balancing technique for the dataset. SMOTE was proposed in 2002 by Chawla et al. [26] as an alternative to class balancing to generate synthetic information.

Because the generation operation is run in attribute space, use of the technique is more general. The SMOTE mode of operation is based on identification of class records to generate synthetic information and the union of these occurrences, equivalent to $k$ close neighbors of the same class, by means of lines. After this step, synthetic occurrences $S_s$ are generated by taking the difference between the vector of attributes $S$, i.e., an occurrence of the class and closest neighbor $S_n$. The difference is multiplied by a random number $r$ in the range 0 to 1 and added to the vector of attributes $S$, resulting in the selection of a random point on the line segments of two specific attributes (2) [26,27]:

$$S_s = S + r \times (S - S_n),\qquad(2)$$

This method resulted in a better accuracy metric for both the training and testing phase, unlike the two aforementioned techniques. It is important to note that this oversampling technique was applied only to the training data and not the test data.

The dataset was split in two subsets resulting in one subset referring to information for use in training, which was equivalent to 80% of the total information; 20% of the information was used during the model testing phase.

### 3.3. Definition of Proposed Model

In this subsection, the model architecture proposed for this research is defined. First is the definition of the model, which is made up of layers of densely connected neural networks. The model has a sequential architecture with seven layers one input, five hidden, and one output.

The initial layer, the input layer, has 78 neurons, which encapsulates the 78 attributes of the dataset used. The hidden layers segment is defined by densely connected layers, which can be understood as those in which each neuron receives an input from all the neurons instantiated in the previous layer [28].

A rectified linear activation function (ReLU) is handled in each layer. ReLU evaluates if the input is greater than 0. If so, the output is the same as the input; otherwise, it is equal to 0. The decision to use ReLU as the main activation function of the model was a consequence of its computational simplicity and its being an evaluation function of the maximum between the input value and 0 [29].

In this model, a layer called Dropout was used. The function of this layer is to ignore a rate of nodes in a random fashion. This in part represents an effect in which neurons can modify the way in which they correct the errors of other neurons, which entails complex co-adaptations that do not generalize unseen information, as raised by Srivastava et al. [30].

The model output layer is instantiated with a Softmax activation function. The choice of this function is associated with the main task of the model, which is classification in a multiclass context. This function will return a probability of the maximum value in an array, which is interpretable as the most probabilistically accurate label for the sample being evaluated [31].

The loss function used in the model is categorical cross-entropy [32], also known as Softmax loss, and the instantiated optimizer is Adam [33]. A representative image of the defined model is in Figure 2.

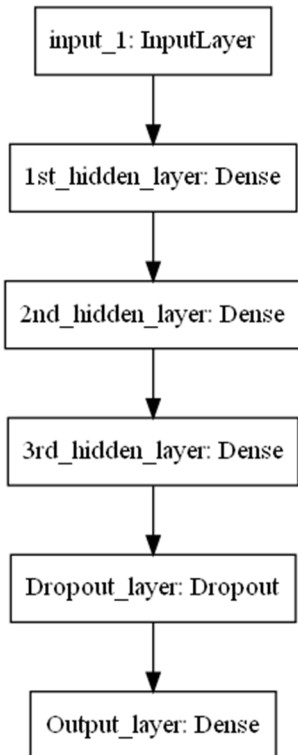

**Figure 2.** Visual representation of defined model.

### 4. Model Evaluation

The information is initially divided for training and testing, with training information constituting 80% of the total information and 20% for testing. Of the 20% of the test information, 10% is used for hyperparameter tuning and the other 10% is treated as previously unseen information for the model.

For the evaluation, we considered three scenarios that focus on the labels used during the training phase. The first scenario considers all the labels in the dataset. The second considers 10 labels that do not present similarity in the attribute space. The third performs a union of labels with strong similarity in the attribute space. The main reason for this separation of scenarios is because of the strong similarity between labels in terms of attributes caused by the information generated by the attack and is captured by the instruments used by Sharafaldin et al. [9].

For all three scenarios, the model structure remains largely constant with a modification in the output layer due to the difference in number of classified labels. The metrics used for evaluation of the hyperparameter selection and evaluation of the scenarios and, in turn, the model metrics, are defined as follows:

$$accuracy = \frac{t_p + t_n}{number\ of\ samples} \tag{3}$$

$$precision = \frac{t_p}{t_p + f_p} \tag{4}$$

$$recall = \frac{t_p}{t_p + f_n} \tag{5}$$

$$f1\ score = 2 \times \frac{precision \times recall}{precision + recall} \tag{6}$$

where $t_p$ = true positives, $t_n$ = true negatives, $f_p$ = false positives and $f_n$ = false negatives.

*4.1. Hyperparameter Tuning*

Firstly, a tuning of the hyperparameters was done to determine the optimal settings for model training. The hyperparameters were configured following a model of configuration, testing, and selection of the best option for the configuration in question. We initially selected the depth of the neural network, i.e., its number of layers. For this choice, a depth in the hidden-layer block of 2, 3 and 4 is proposed. Based on the result, a depth of 3 was chosen, owing to model accuracy and the time spent in its training.

The number of neurons instantiated in each layer was configured based on the choice of best configuration candidate. During this procedure, about 10 different configurations were evaluated, and the structure (128,256,512) was chosen, which refers to the number of neurons of the 3 layers previously configured and belonging to the block of hidden layers.

The learning rate of the Adam optimizer is configured based on three candidate rates, $1 \times 10^{-4}$, $1 \times 10^{-5}$, and $1 \times 10^{-6}$. Table 2 shows the results according to the learning rate used.

**Table 2.** Performance evaluation for the training rate of the model.

| Rate | Accuracy | Precision | Recall | F1 Score |
|------|----------|-----------|--------|----------|
| 0.001 | 0.9370 | 0.9389 | 0.9352 | 0.9370 |
| 0.0001 | 0.9421 | 0.9421 | 0.9403 | 0.9412 |
| 0.00001 | 0.9379 | 0.9400 | 0.9359 | 0.9379 |

Based on this test, we selected the learning rate corresponding to $1 \times 10^{-4}$ for configuration of the Adam optimizer. As defined above, the dropout layer contains a rate of total number of neurons to ignore.

For this model, we discovered that the rate of 0.3 gave the best tradeoff between time of convergence of the model and its accuracy. Table 3 compares the rates used for configuration of the Dropout layer. The training time in minutes (TTM) column represents the maximum time that the accuracy metric did not vary by more than 0.0001 units.

**Table 3.** Performance evaluation of the different rates for dropout layer.

| Rate | Accuracy | TTM | Recall | F1 Score |
|------|----------|-----|--------|----------|
| 0.20 | 0.9259 | 25 | 0.9255 | 0.9260 |
| 0.30 | 0.9421 | 30 | 0.9403 | 0.9412 |
| 0.35 | 0.9366 | 33 | 0.9299 | 09360 |

*4.2. Results of the Model*

Before presenting the results of the model, Table 4 indicates the hyperparameters chosen for the model. Hyperparameters like the number of epochs, batch size and neurons per layer were configured using the same approach as the others hyper-parameters. In other words, a set of configurations was tested, and the best performing setting was chosen.

**Table 4.** Hyperparameter settings for the proposed model.

| Hyperparameter | Configuration |
|----------------|---------------|
| Number of layers | 3 |
| Adam optimizer learning rate | $1 \times 10^{-4}$ |
| Dropout rate | 0.30 |
| Epochs | 100 |
| Batch size | 128 |
| Neurons per layer (layer #1 through layer #3) | 128,256,512 |

In order to evaluate our model, we proposed three different scenarios. Each scenario has a different experiment configuration. The differences are within the number of labels.

The reason behind this methodology is to see how the model performs under different condition and to evaluate which would be the best condition in which the model would perform optimally.

The results presented were obtained via cross validation. The setting for the cross validation was a 5-fold validation for scores such as: accuracy, precision, recall and F1 score. These settings were maintained throughout the three different scenarios.

### 4.2.1. First Scenario

For this scenario, the model was trained to categorize 13 labels belonging to the dataset, equivalent to those shown in Table 1. The model evaluation for this scenario resulted in the following values for the metrics of accuracy, precision, recall, and F1 score presented in Table 5.

**Table 5.** Model training results for the first scenario.

| Accuracy | Precision | Recall | F1 Score |
|----------|-----------|--------|----------|
| 0.8177 | 0.831 | 0.7995 | 0.8149 |

The confusion matrix resulting from the test is displayed in Figure 3. This matrix provides more detailed information on the labels, for which the model had a precision less than the average reflected in Table 5.

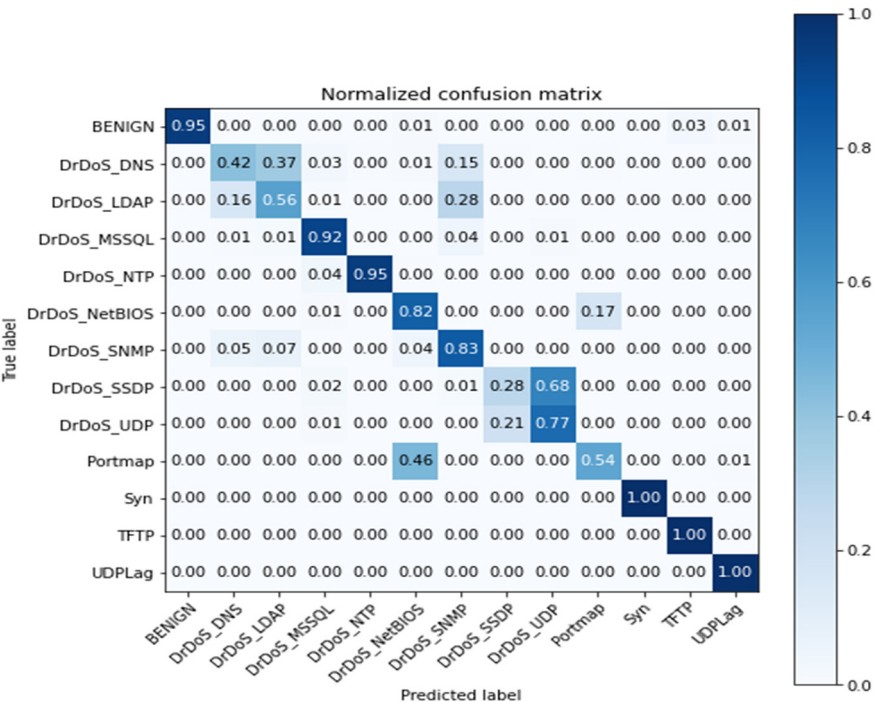

**Figure 3.** Confusion matrix for the first scenario.

In this scenario, the model exhibits confusion for the DrDoS DNS-DrDoS LDAP labels. This confusion lies in the nature of the attacks and their similarity, as the information captured by the network monitoring instrument is similar. The values reflected in the dataset attributes will also have this similarity, thereby negatively affecting the metrics of the model. The same behavior is observed for the Portmap, UDP, SSDP, and NetBios labels.

### 4.2.2. Second Scenario

For this scenario, the model was trained for the categorization of 10 labels: BENIGN, DNS, MSSQL, NTP, NETBIOS, SNMP, SSDP, SYN, UDP-LAG, and TFTP. The attacks such

as Portmap, UDP, and LDAP were removed from the testing subset, the training was made using the same labels as scenario 1. Accuracy, precision, recall and F1 score metrics are listed in Table 6. This is the result of model training in this scenario.

**Table 6.** Model training results for the second scenario.

| Accuracy | Precision | Recall | F1 Score |
|----------|-----------|--------|----------|
| 0.9457 | 0.9475 | 0.9438 | 0.9456 |

The confusion matrix of the model evaluation stage is shown in Figure 4.

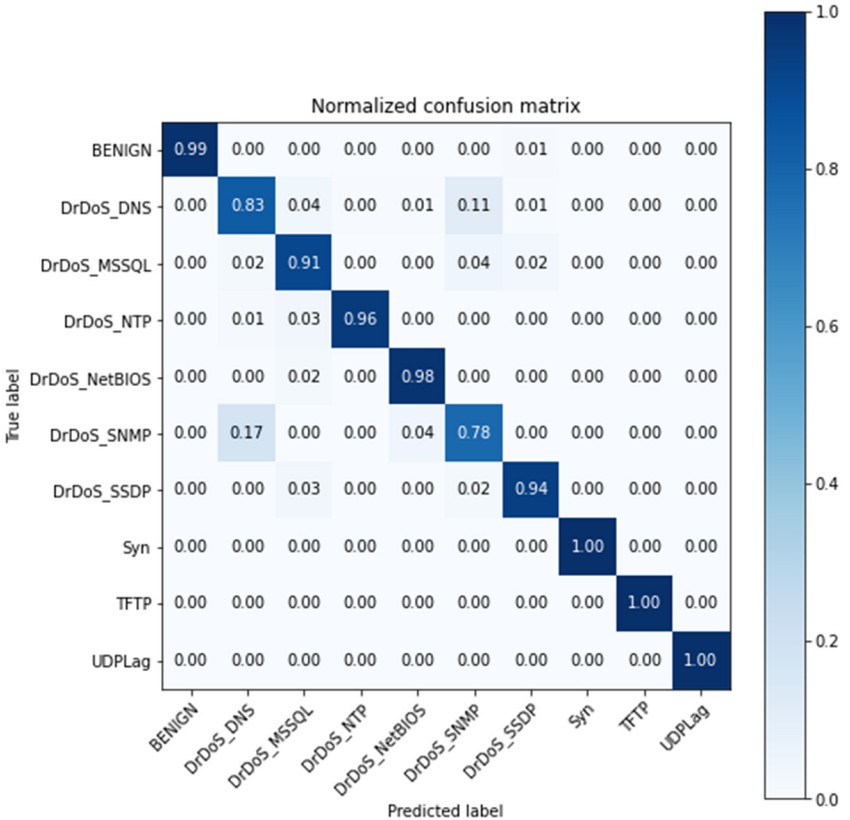

**Figure 4.** Confusion matrix for the second scenario.

This matrix reveals a better result compared to Figure 3, owing to the non-consideration of labels with values that are over-lapped in the attribute space. Although the scores in this scenario are good, this scenario is mainly deployed to have a better idea on whether the model is finding a high similarity in between the labels.

4.2.3. Third Scenario

In the development of the third scenario, it was decided to unify the attacks with strong similarity in the attribute space. These new labels are composed of the names of the attacks that show this similarity. The new labels are DrDoS DNS/LDAP, DrDoS NetBIOS/Portmap, and DrDoS SSDP/UDP. The model was retrained under the newly created mixed labels. Also, the testing subset possess the newly created labels instead of the previous labels, bringing the total of labels to 10 and still maintaining the same number of attacks. Table 7 demonstrates the similarity between the labels in attribute space. The name of the subset is equivalent to the union of labels made; this subset contains only the two labels that show strong similarity.

**Table 7.** Summary of subsets created for labels with strong similarity.

| Subset | ACV | AFR0, % | ASAH, % |
|---|---|---|---|
| DNS/LDAP | 12 | 24, 99.9 | 38, 96.8 |
| NetBIOS/Portmap | 12 | 31, 98.0 | 41, 98.3 |
| SSDP/UDP | 12 | 25, 99.9 | 32, 97.9 |

The abbreviation ACV refers to attributes with constant value, AFR0 encapsulates the attributes with a high frequency ($\geq$98%) of zeros, and ASAH contains the attributes that have a strong bias due to a strong homogeneity of their values ($\geq$96%). AFR0 and ASAH also contain the respective percentages according to the metric.

The results of training of this model are shown in Table 8.

**Table 8.** Model training results for the third scenario.

| Accuracy | Precision | Recall | F1 Score |
|---|---|---|---|
| 0.9421 | 0.9421 | 0.9403 | 0.9412 |

The resulting confusion matrix from the testing phase is presented in Figure 5.

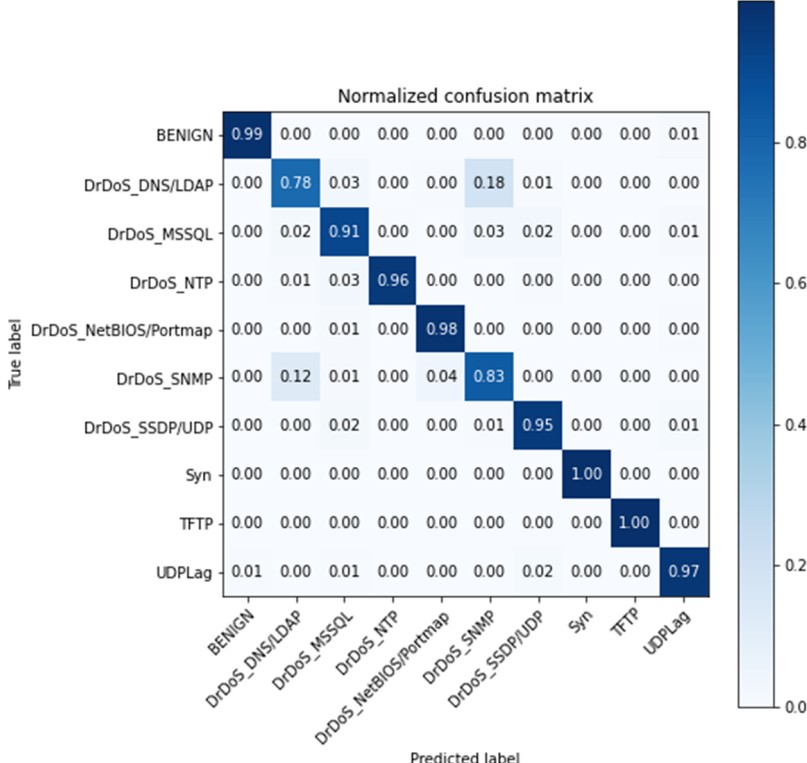

**Figure 5.** Confusion matrix for the third scenario.

## 5. Discussion

The main contribution of this work is a framework for data preprocessing and a classification model in a multiclass context for DDoS attacks. Compared to previously reviewed proposals in the literature, in addition to categorizing the flow as malignant or benign, the present model provides the type of attack evidenced in the flow (if it is malignant). This characteristic is relevant when developing a security strategy for deployment in a network that has experienced or is at risk of a DDoS attack.

The proposed model for the categorization of DDoS attacks can be deployed in the application layer within an SDN software-defined network, as proposed by Sharafaldin et al. [9]

and based on the implementation in Alshamrani et al. [34]. The aim is to be able to block malignant flows or route them to analyze the flow and feed the future training of the model and increase its stability [35].

A summary table stating the contribution of this paper, approach, dataset, limitation and metrics related to the proposed model is presented in Table 9.

**Table 9.** Result comparison with related works.

| Author [Reference] | Approach | Average (Precision, Recall, F1 Score) |
|---|---|---|
| Can et al. [36] | 82 features + FS | 91.16, 79.41, 79.43 |
| Ferrang et al. [37] * | DNN | 70.00 |
| | RNN | 70.00 |
| | CNN | 66.46 |
| Sharafaldin et al. [9] | Naive Bayes | 30.30, 17.51, 7.35 |
| | SVM | 62.44, 57.97, 55.50 |
| | Decision Tree | 61.15, 58.32, 55.15 |
| | Random Forest | 50.76, 36.91, 39.57 |
| Our approach | DNN | 94.21, 94.03, 94.12 |

\* No distinction for the metrics is stated by the author.

In Table 9, we can see how our approach outperforms those presented by Can et al. [36] Ferrang et al. [37] and Sharafaldin et al. [9], the latter being the benchmark. We attribute the scores achieved by our model due to processing phase of the research and the combined labels proposed previously.

Our proposed model is limited by the labels used in the training phase, meaning, it will not be able to detect accurately whether a new attack is not present before is detected in a malign flow. In the case an attack is present in a flow set to be evaluated by the model, the model will classify as malign, but it will output an incorrect type of attack. This can be solved by providing a new label during the training phase with the name NEW. This will encapsulate a new attack and reduce the ambiguity of the output.

## 6. Conclusions and Future Work

In this work, a preprocessing stage and multiclass classification neural network model are presented. During the preprocessing stage, various transformation techniques were applied to the dataset. This improved the accuracy metrics and training time of the proposed model.

Different datasets were evaluated for model training, and it was decided to conduct this work with the dataset provided by researchers from the CIC. During the proposed preprocessing stage, it was possible to perform a cleaning of NaN-type and null values. Similarly, a data normalization and quantile transformation were performed on the dataset. To obtain a balanced dataset in the context of classes, the SMOTE technique was applied only to the sample used for model training.

The model was tuned based on a hyperparameter configuration. From the results of model training and validation, the tuning was applied mainly to the number of hidden layers, the rate used in the Dropout layer, and the learning rate. The model was trained and evaluated in three different scenarios. The results, metrics, and confusion matrix were presented for each scenario.

The results show the third scenario as the best. This analysis was based on the performance metrics of the scenarios and number of labels used in the training phase. Scenario 2 yielded a 94.57% accuracy, greater than Scenario 3 by 0.31%. However, it was not trained with the 13 labels available in the dataset, unlike the third scenario, which was trained with all labels. The added value of the proposed model consists in identifying the type of attack evidenced in a flow, as compared with the binary classification of recent works.



As future work, we propose to deploy and evaluate the model in a computer network that has a flow not previously seen by the model. This deployment will consist of a model evaluation phase. Depending on the results, we propose to conduct a model training phase with the captured data. Once the model has been trained and validated, it could be instantiated as a network traffic manager whose objective is to reject or accept network flows based on the evaluation performed. Furthermore, this could be deployed in an intrusion-tolerant-system in order to prevent denial of service attacks, as proposed by Kwon et al. [38].

**Author Contributions:** Conceptualization, A.C. and J.M.; methodology, A.C. and J.M.; software, A.C.; validation, A.C. and J.M.; formal analysis, A.C.; investigation, A.C.; resources, A.C.; data curation, A.C.; writing—original draft preparation, A.C. and J.M.; writing—review and editing, J.M.; visualization, A.C. and J.M.; supervision, J.M.; project administration, J.M. All authors have read and agreed to the published version of the manuscript.

**Funding:** This research received no external funding.

**Institutional Review Board Statement:** Not applicable.

**Informed Consent Statement:** Not applicable.

**Data Availability Statement:** The data used in this study are openly available in Canadian Institute for Cybersecurity at https://www.unb.ca/cic/datasets/ddos-2019.html (accessed on 12 January 2021).

**Acknowledgments:** Our highest recognition for the Systems Engineering department of the Universidad del Norte.

**Conflicts of Interest:** The authors declare no conflict of interest.

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
