# Peer review of "Multi-Classifier of DDoS Attacks in Computer Networks Built on Neural Networks"

_applsci, doi:10.3390/app112210609_

Round 1

Reviewer 1 Report

The authors seek to propose a Machine Learning model for effective network traffic classification for possible DDoS attacks. The following observations were, however, made in the manuscript.

  1. The authors do not highlight the limitations of existing works. Identification of limitations of state-of-the-art works can provide justification and relevance for undertaking this research.
  2. To aid easy perusal, the authors need to provide a summary table highlighting the contributions, approach, dataset, limitations, accuracy, etc.
  3. Due to the rapidly changing nature of networks, the authors need to justify why they used the CIC dataset instead of acquiring their dataset to reflect current network situations adequately.
  4. The authors should provide the model plot instead of the current Fig.2. It will provide a more informative representation of the proposed model.
  5. The authors need to tabulate the hyper-parameters used and their corresponding values during their proposed model's training phase.
  6. Results provided in table 2 shows that 1e-4 outperformed 1e-5 but the authors chose 1e-5 as the best hyper-parameter
  7. The authors present results for three scenarios but do not provide clear justifications for these scenarios. They need to provide clear reasons for undertaking such activity.
  8. In the abstract, the authors claim, "The result was compared with previous works treating the same dataset used herein." However, nothing is seen in support of that claim in the main manuscript.
  9. The authors need to compare their results to other related works that were also carried out on the CIC dataset (e.g., https://doi.org/10.3390/electronics10111257, etc.). This can help in validating the relevance of their proposed model instead of making self-comparison.

Author Response

Thank you for your feedback.

Reviewer 2 Report

The research proposed a multi–classifier Neural Network framework for detecting DDoS attacks, evaluated by three scenarios.

Comments:

  1. Abstract: The dataset's name should be included in the abstract. Summarize the most important numerical results from the experiments.
  2. On page 1, line 37 stated that: "There has been an increase in the number of such attacks in the last ten years…."  by reference [3],  a research from 2004?
  3. Please give some references for the paragraph on lines 50 to 55.
  4. This study's novelty and contribution are both too brief and unclear. Please state the novelty and contribution at the end of the introduction.
  5. In line 108, you are speaking about some previous works " The author highlighted the importance of the processes….". I recommend providing the impacts of processing briefly.
  6. From line 131 to line 152, I am confused about the mixing presentation of the research results of [10] and [11]. Please rewrite it more coherently.
  7. I do not consider paragraphs from lines 167 to 174 beneficial. It is pretty apparent when not using outdated data sets. Consider removing it.
  8. 1 is blurred. Please improve the quality.
  9. In line 215 said, "… we perform a new retrieval of the files in CSV format.". Please describe this in detail.
  10. Please describe the model architecture in the paragraph from line 295 to line 321 in a table structure.
  11. Please describe the second scenario in more detail. How about the other labels not in categorization, you remove them or process them in some way? Is it reasonable to compare the second scenario results to other scenarios where the data set is no longer equivalent if you remove them?
  12. I see in Table 2 the best result for Adam hyperparameter tuning is 1x10^-4 .
  13. In the "Hyperparameter Tuning" section, please explain how data is organized and followed which scenario ( or not ).
  14. Please explain more detail about cross-validation results in the article.
  15. Please add the discussion section and discuss the limitations of the proposed approach and the threats-to-validity of the experimental results.
  16. There are many malformed references such as [10] ,[11] ,[14] , add journal or conference information, and double-check all references.

Author Response

Thank you for your feedback.

Reviewer 3 Report

This paper deals with an exciting topic. The article has been read carefully, and some crucial issues have been highlighted in order to be considered by the author(s).

All the acronyms should be defined and explained first before using them such that they become evident for the readers.

The Introduction and related work parts give valuable information for the readers as well as researchers. 

In addition recent papers should be added in the part of related work.

Representation of figures needs to be improved.

It would be good if DDoS security domains [1], such as Intrusion tolerant system, would be reflected in future research or related work.

[1] Kwon, Hyun, Yongchul Kim, Hyunsoo Yoon, and Daeseon Choi. "Optimal cluster expansion-based intrusion tolerant system to prevent denial of service attacks." Applied Sciences 7, no. 11 (2017): 1186.

Author Response

Thank you for your feedback.

Round 2

Reviewer 1 Report

  1. The limitation identified by the authors for state-of-the-art models as being binary classification does not hold. The example cited below used multi-class classification on the same dataset as the authors. Can, D.C., Le, H.Q. and Ha, Q.T., 2021, April. Detection of Distributed Denial of Service Attacks using Automatic Feature Selection with Enhancement for Imbalance Dataset. In Asian Conference on Intelligent Information and Database Systems (pp. 386-398). Springer, Cham.
  2. My earlier comment about the summarized table as shown in Table 9 was supposed to be for the related works reviewed by the authors and not the work being proposed by the authors. That would have easily convinced readers to appreciate the current terrain and scope of work carried out in the domain of DDoS attack classification.
  3. The authors do not still provide justification on the use of the CIC DDoS attack 2019 instead of a more current one due to the rapidly changing nature of networks.
  4. The training rate in Table 4 should be the "learning rate". The authors did not indicate the number of epochs, optimizer, loss etc.
  5. The authors do not still compare their results with existing works for validation purposes. They only provided a sentence that claims that their results are comparable to Ferrang et al. There are other related works they could have also compared with. The authors are to plot their results with those from existing works for better clarity. The authors can add that of Can et al. (cited above) but are not limited to these two.

Author Response

Thank you for your feedback.

Reviewer 2 Report

In point 13 of the previous review, in the "Hyperparameter Tuning" section of the manuscript, please elucidate how many labels the model was trained to classify during the tuning process. Is this have any effect on the outcomes of the three proposed scenarios?

Author Response

Thank you for your feedback and answer to our question regardless your comment in the first revision. 

Reviewer 3 Report

This paper is worth for acceptance.

Author Response

Thank you for your feedback. 

Round 3

Reviewer 1 Report

The author has tackled all raised concerns